# A Face Detection and Standardized Mask-Wearing Recognition Algorithm

**DOI:** 10.3390/s23104612

**Published:** 2023-05-10

**Authors:** Jimin Yu, Xin Zhang, Tao Wu, Huilan Pan, Wei Zhang

**Affiliations:** 1College of Automation, Chongqing University of Post and Telecommunications, Chongqing 400065, China; yujm@cqupt.edu.cn (J.Y.); S200303029@stu.cqupt.edu.cn (T.W.); zhangwnsw@163.com (W.Z.); 2School of Science, Chongqing University of Posts and Telecommunications, Chongqing 400065, China; panhl@cqupt.edu.cn

**Keywords:** wearing mask detection, feature pyramidal, bounding box regression, confidence

## Abstract

In the era of coronavirus disease (COVID-19), wearing a mask could effectively protect people from the risk of infection and largely reduce transmission in public places. To prevent the spread of the virus, instruments are needed in public places to monitor whether people are wearing masks, which has higher requirements for the accuracy and speed of detection algorithms. To meet the demand for high accuracy and real-time monitoring, we propose a single-stage approach based on YOLOv4 to identify the face and whether to regulate the wearing of masks. In this approach, we propose a new feature pyramidal network based on the attention mechanism to reduce the loss of object information that can be caused by sampling and pooling in convolutional neural networks. The network is able to deeply mine the feature map for spatial and communication factors, and the multi-scale feature fusion makes the feature map equipped with location and semantic information. Based on the complete intersection over union (CIoU), a penalty function based on the norm is proposed to improve positioning accuracy, which is more accurate at the detection of small objects; the new bounding box regression function is called Norm CIoU (NCIoU). This function is applicable to various object-detection bounding box regression tasks. A combination of the two functions to calculate the confidence loss is used to mitigate the problem of the algorithm bias towards determinating no objects in the image. Moreover, we provide a dataset for recognizing faces and masks (RFM) that includes 12,133 realistic images. The dataset contains three categories: face, standardized mask and non-standardized mask. Experiments conducted on the dataset demonstrate that the proposed approach achieves mAP@.5:.95 69.70% and AP75 73.80%, outperforming the compared methods.

## 1. Introduction

Since the discovery of the novel coronavirus that caused COVID-19 in December 2019, the outbreak has spread to various counries. As of 1 April 2023 CET, 761,402,282 COVID-19 infections have been reported by the WHO worldwide, with 6,887,000 deaths [1]. To prevent the wide spread of the virus, the international community has used vaccinations, masks, area closures, virus testing and cutting off the source of the virus to combat the epidemic. These methods have somewhat reduced the impact of the virus on human economic, social and physical health. It has become an international consensus that wearing masks is the easiest and most effective way to prevent infection, cut off the chain of transmission, and prevent the spread of the epidemic [2].

To combat various infectious viruses, governments need to direct prevention and control in public places and assign monitors, for example, non-contact temperature measurements through monitoring instruments [3,4,5]. However, monitoring large numbers of people in most places is a challenging task that involves detecting the wearing of masks, which most monitoring instruments lack. However, this can be achieved through integration between monitoring devices and deep learning techniques.

Traditional methods usually used manual features for face detection. One of the most used features is a Haar-like feature, which can be trained by the AdaBoost algorithm for face detection [6]. Dewantara and Rhamadhaningrum [7] exploited the AdaBoost algorithm with Haar, LBP, and HOG features to train a cascade classifier for multipose masked face detection. It is reported that using the Haar-like feature achieves a higher accuracy. Petrovic and Kocic [8] introduced an affordable IoT-based system for COVID-19 indoor safety. The mask detection method is based on three libraries in OpenCV: frontal face, mouth and nose classifiers. It detects the face first, and then verifies it using the characteristic of mouth and nose.

With the continuous mutation and spread of viruses, deep learning-based mask detection is becoming a hot area of interest. To obtain a highly accurate algorithm, Loey et al. [9] proposed a hybrid deep migration learning model and machine learning method for masked face detection. It uses ResNet-50 [10] to extract feature maps and employs decision trees, supports vector machines and integrated algorithms for recognition, but the monitoring speed is greatly reduced while obtaining high accuracy. Wang et al. [11] used the spatial and frequency features from the 3D information to detect masks. Compared with the YOLOv4 [12] algorithm, which has a high balance of accuracy and speed, it has a faster detection speed but worse detection accuracy. Qin et al. combined image super-resolution and classification networks as new conditions for identification of mask-wearing. Experimental results indicate that the addition of image super-resolution can improve the classification accuracy by 1.5% compared to the deep learning method without a super-resolution module. Accurate locations of facial masks can improve the accuracy of face recognition algorithms [13,14,15]. Each of these methods has advantages in terms of speed or accuracy, depending on the application requirements. Sui et al. [16] proposed a non-reciprocity sensor based on a layered structure with multitasking. This instrument can distinguish between normal cells and certain abnormal cells. Combined with facial and mask detection algorithms, it is expected to detect whether the mask has been properly worn while also detecting whether it has been infected with related viruses. Some algorithms [17,18,19] have helped in balancing speed and accuracy, but overall performance still needs to be improved.

We have been interested to monitoring whether people in public places wear masks or not and propose a face and mask detection algorithm with high accuracy and real-time balance. To get better detection performance, we consider object detection based on YOLOv4 which has higher performance and will achieve better results.The contributions of this paper are as follows:•Committed to getting a higher accuracy of the bounding box regression, and based on CIoU [20], we propose a new function which is called Norm CIoU(NCIoU). We add a penalty term based on the L1 and L2 norm, which carries the bounding box information of the object, which aims to help more prediction boxes accurately return.•The Binary Cross Entropy (BCE) function and the Mean Square Error (MSE) function are used to calculate the confidence loss when there is an object and when there is no object. Since most of the area of the image is background information, MSE will not provide an excessive loss value when calculating the confidence loss when there are no objects, which alleviates the tendency of the model to be biased towards no objects.•To compensate and reduce the loss of object information during the feature fusion process, the cross stage lateral connections are added based on a Path Aggregation Network (PANet) [21] to compensate for the object information loss caused by the convolution and pooling operations. At the same time, to highlight the object information to reduce the loss of target information in the feature fusion, the hybrid attention mechanism HAMNet [22] is introduced to highlight the object information and suppress the expression of background information, thus proposing a novel feature pyramid network based on an attention module.

We select YOLOv4 as the basic model to apply our methods for better performance. In Section 2, we discuss related work. In Section 3, detailed introduction for our proposed method will be presented. In Section 4, we analyze and present the experimental results. Finally, conclusions are drawn in Section 5.

## 2. Related Work

Driven by its high accuracy, low cost and acceptability object algorithms are used for face and mask-wearing recognition. We are committed to proposing some methods to improve the accuracy of the model and reduce the probability of false detection and missed detection, to be more widely used in reality. We will briefly introduce the work related to our research content in three parts: feature fusion network, bounding box regression loss function and confidence loss function.

### 2.1. Feature Fusion Network

Feature fusion fuses different feature maps obtained from different feature extractors and has great potential to achieve better classification performance. Most of the many feature fusion networks use only deep-level features, and it is difficult to accurately preserve the location features of the object due to their low resolution. To obtain the location information in the shallow feature maps, Feature Pyramid Network (FPN) [23] is proposed to solve the problem of multi-scale feature fusion. Several widely used feature fusion networks are shown in Figure 1.

In Figure 1a, the feature diagram obtained a variety of scale feature diagrams after multiple samples to form an image pyramid. This method has achieved good results in multiple datasets, but the calculation of this method is very large. The method is shown in Figure 1b only uses a single feature map to predict, but the shallow feature map contains more location information, and deep features include semantic information, which is very unfavorable for object detection. The method shown in Figure 1c using the features of different resolutions to perform the object location prediction, which can handle multi-scale problems more effectively, but the resolution of the underlying feature map is insufficient, which is very unfavorable for the detection of the small object. Figure 1d is the FPN. The horizontally connected feature fusion architecture makes the characteristic diagram of all sizes have high-level semantic information. The authors applied it to Faster R-CNN [24], and compared to the original network AP50 improved 9.6% on COCO minival set.

As shown in Figure 2, PANet is a further improvement of FPN, which increases the multi-path weighted aggregation layer, so that the model can obtain multi-scale semantic characteristics of different layers on a weight, thereby bringing the accuracy and stability of the detection. At the same time, PANet uses path aggregation modules to enhance the ability of the network, and enhances the characteristics of other layers according to the characteristics of different layers, so as to achieve stronger feature aggregation. The AP50 of PAnet on Mask R-CNN [25] is 4.9% higher than FPN.

In these feature fusion networks, the top-down feature fusion path enables the deep feature map to contain the position information of the object. The bottom-up feature fusion path makes the low-level feature image contain semantic information. Based on such an idea, we worked on increasing the object information of the final feature map.

### 2.2. Bounding Box Regression

Traditional bounding box regression loss functions often take the form of L1 or L2, where the L1 function can provide stable gradients during training to improve location accuracy using L1, which can enhance the detection effect at a given scale. The L2 function can produce smoother regression results but also lacks sufficient robustness to the presence of outliers. To address these issues, researchers have proposed some improved loss functions for bounding box regression such as the SmoothL1 [24] loss function and the IoU [26] loss function. The Smooth L1 loss function is a loss function that balances smoothness and robustness, and A 69.9% mAP was obtained on PASCAL VOC 2007. It can effectively suppress the effect of outliers in the marginal regression process. Its expression is given in Equation (Equation 1).
(1)$${Smooth}_{L_{1}}=\left\{\begin{array}{cc} 0.5x^{2} & |x|<1\\ |x|-0.5 & \;\mathrm{others}\; \end{array}\right.$$

Since the variation of the real box size can lead to a large variation of the gradient of the loss function, problems such as gradient explosion can occur. In contrast, the Smooth L1 function uses a quadratic function in the small parameter region and a linear function in the large parameter region, which can maintain better stability when the object box size varies widely. The gradient is relatively small when the prediction box deviates far from the real box, which equivalently means that there is no operation that needs to be corrected. This property is equivalent to a regularisation term, which helps to alleviate the problem of overfitting and also makes the model converge more easily to the extreme value.

On the basis of GIoU [27] and DIoU [20], CIoU takes into account the overlap area, the distance of center points between real box and prediction box, and the aspect ratio. Applying the loss function on YOLOv3 and conducting experiments on PASCAL VOC 2007, the AP75 of CIoU is 9.88% and 9.31% higher than IoU and GIoU, respectively. In the object detection process, scale differences and aspect ratio differences between prediction box and real box are common problems. CIoU loss function takes these differences into account by introducing scale and aspect ratio views to make the training more stable. The geometry and position information of the object box are also considered, thus better capturing the similarity of the object box. The formula can be stated as:(2)$$CIoU=IoU+\frac{p^{2}\left(b,b^{gt}\right)}{c^{2}}+\alpha v$$

Among them, IoU is the intersection ratio of the prediction box and the real box, respectively, and $p^{2}\left(b,b^{gt}\right)$ represents the euclidean distance between the center points of the two boxes. *c* represents the diagonal distance of the smallest closed area that can contain both the prediction box and the ground truth box. α is a parameter function used to balance the ratio, *v* is used to evaluate the consistency of the ratio between the two boxes. Their formulas are as follows:(3)$$\alpha =\frac{v}{1-IoU+v}$$
(4)$$v=\frac{4}{\pi }{\left(\tan^{-1}\frac{w^{gt}}{h^{gt}}-\tan^{-1}\frac{w}{h}\right)}^{2}$$
where *w* and *h* denote the width and height of the prediction box, respectively, and $w^{gt}$ and $h^{gt}$ denote the width and height of the real box, respectively. The performance improvement process of the border regression function obtains higher accuracy by adding more information about the bounding box to measure the loss in a suitable form. This is still the mainstream approach to improving the performance of functions.

### 2.3. Confidence Loss Function

Confidence is used to determine whether the object inside the bounding box is a positive or negative sample, and early object detection methods such as YOLOv1 [28] and YOLOv2 [29] usually use MSE function to train the model. The MSE loss function is more sensitive in handling outlier samples and is susceptible to samples with large errors, leading to performance degradation. SSD [30] introduces the Softmax function as the confidence loss function. This loss function treats the confidence of each detection box as the probability of the object class, which is normalized to reflect whether the prediction box contains the object or not. The core idea of Focal loss [31] is to solve the category imbalance problem by reducing the weight of easy-to-classify background samples and increasing the weight of hard-to-classify foreground samples, which can significantly improve the detection performance of models on highly imbalanced datasets.

BCE function is a widely used confidence loss function in object detection tasks and has now been shown to perform well in some classical algorithms [12,32,33,34]. Its form is as follows:(5)Loss=ylog(x)+(1−y)log(1−x)

The YOLOv4 algorithm uses BCE as a confidence loss function, which treats confidence calculation as a binary classification problem. The closer the confidence prediction value is to 1, the more likely it is that the current grid has targets, the less likely it is otherwise. If you simply use 0 and 1 as confidence labels, when the grid has objects, the output is 1, and vice versa. The form of the confidence loss function for YOLOv4 is as shown in (Equation 6). The picture is divided into multiple grids, and each grid with or without objects is calculated using BCE, so the function is divided into two parts. However, the two parts of this function do not participate in the calculation simultaneously, as each grid will only have objects or no objects. Whether or not to participate in the calculation is determined by $I_{ij}^{obj}$ and $I_{ij}^{noobj}$. If a grid contain objects, then $I_{ij}^{obj}$ = 1, $I_{ij}^{noobj}$ = 0, otherwise $I_{ij}^{obj}$ = 0, $I_{ij}^{noobj}$ = 1.
(6)$${\mathrm{loss}}_{\mathrm{conf}\;}=\sum_{i=0}^{S^{2}}\sum_{j=0}^{B}\left(I_{ij}^{\mathrm{obj}\;}+\lambda_{\mathrm{noobj}\;}I_{ij}^{\mathrm{noobj}\;}\right)\left[C_{i}^{j}\log\left({\hat{C}}_{i}^{j}\right)+\left(1-C_{i}^{j}\right)\log\left(1-{\hat{C}}_{i}^{j}\right)\right]$$
where $S^{2}$ is the number of grids divided by each image, and *B* is the prior box of each grid generated by the network. $\lambda_{\mathrm{noobj}\;}$ is the weight of loss when the image without objects. $C_{i}^{j}$ indicates whether there is an object in the i-th grid. If there is an object, $C_{i}^{j}$ = 1; otherwise, it is 0. ${\hat{C}}_{i}^{j}$ represents the probability of predicting the existence of the object, and the value is [0, 1].

The BCE function can help with better regression of the bounding box, but it can greatly amplify the loss of negative samples and affect the accuracy of the model. The MSE function does not amplify the loss of negative samples and can reduce the interference of noise.

### 2.4. Hybrid Attention Module

In the process of feature fusion from images using deep learning techniques, the contribution of different regions of the image to the model task is unbalanced. The task-relevant regions contain the key information that the model needs to focus on. Spatial attention considers only spatial feature information and treats features of different channels in the same way, resulting in information interactions between different channels being ignored. Similarly, channel attention considers only channel feature information and performs the same processing on spatial features within the same channel, resulting in ignoring information interactions between different spatial features. Hybrid attention consists of both and takes full advantage of both.

HAMNet is a more lightweight hybrid attention mechanism compared to CBAM [35]. Using HAM on ResNeXt29, the CIFAR-10 error and CIFAR-100 error were reduced by 0.62% and 1.15%, respectively, and were 0.46% and 0.89% lower than using CBAM, respectively. In HAMNet, a channel feature map and channel refinement features are first generated after the channel attention module. After the feature map is output by the channel attention submodule, the spatial attention submodule divides the input feature map into two groups to generate a pair of spatial attention descriptors, after which the spatial sub-module generates the final refinement features that can be adaptively adjusted to highlight the target by applying the initial attention descriptors. The channel attention sub-module is shown in Figure 3.

The average pooling and maximum pooling functions perform different functions at different stages of image feature extraction, using an adaptive mechanism to make the weights of the two poolings different. While capturing the association of cross-dimensional information in the same way as ECANet [36]. After passing through the channel attention sub-module, the feature map will generate two different feature tensors, $P_{C}^{\max}$ and $P_{C}^{\mathrm{avg}}$, representing the maximum feature set and the average feature set, respectively. Both tensors are then input into the adaptive mechanism to obtain rich features $P_{C}^{add}\in R^{1\times R\times R}$. It uses a one-dimensional convolution mechanism to solve the problem of channel dimensionality reduction brought about by multi-layer perceptrons, avoiding the increase in model complexity. At the same time, it can model the interaction between channels, use activation functions to compress the output into the range of [0, 1]. Then use the output as the input of the next layer.

In the spatial attention module, the channel separation operation divides the channel refinement features into two groups along the channel axis. A hyperparameter that controls the separation rate is the boundary between important and sub important channel groups. This hyperparameter is multiplied by the channel dimension of the channel refinement feature to obtain the dimension of the important channel features. The spatial attention sub-module is shown in Figure 4.

The spatial channel module pools both the important feature group $P_{1}^{{}^{\prime }}$ and the sub important feature group $P_{2}^{{}^{\prime }}$ along the channel axis direction and generates corresponding feature maps. In addition, a pair of spatial attention feature maps are generated using shared convolutions. This pair of spatial feature maps will undergo a series of non-linear steps to generate spatial attention $A_{s,1},A_{s,2}\in R^{H\times W}$. The sequence is: batch normalization, ReLU [37] activation function, and Sigmoid activation function. This can eliminate negative spatial elements so that attention is focused only on valid feature information.

To fuse multi-scale feature maps, practices such as upsampling and downsampling can lose some valid information. The attention module enhances the model’s focus on objects, thus reducing the loss of object information. The lightweight attention module HAM works in feature fusion networks without increasing the complexity of the network. This helps to build networks with balanced speed and accuracy.

## 3. Proposed Method

In this section, to improve the performance of the bounding box regression loss function, a new penalty term is introduced in detail, which carries the box information and has some advantages of L1 and L2 norms. A more effective feature fusion network AM-NFPN and the confidence function for combining BCE and MSE will also be introduced in detail.

### 3.1. Bounding Box Regression Function

Bounding box regression is one of the key steps in object detection algorithms. The bounding box regression loss function is an important research point, problems such as convergence speed, gradient disappearance, gradient explosion and learning instability still need to be improved. In general, the more information contained in the loss function between the prediction box and the real box, the more accurate the box loss function can be to shorten the distance between the two boxes. Small deviations in regression parameters can lead to large IoU errors for small objects.

As shown in Figure 5, the red box represents the real box, and the blue box represents the prediction box. Given a fixed offset of 5 for the regression parameter, the smaller the object size, the smaller the IoU. Therefore, positioning accuracy requires that small objects be more stringent than general objects. To meet the detection of small objects such as masks, we propose a new bounding box regression loss function to improve the accuracy of prediction box in face and mask wearing detection tasks from the perspective of fusing more information about box.

#### 3.1.1. Basic Penalty

Each step of improvement from IoU to CIoU is based on adding a penalty item of box related information to the IoU loss function to improve the performance of the original function. GIoU adds a penalty item that focuses on the positions of two boxes on the basis of IoU to solve the problem when the box does not overlap. On the basis of IoU, DIoU adds the penalty of the euclidean distance of the center point of the box to improve the convergence accuracy and speed. Based on DIoU, CIoU adds the penalty of the aspect ratio of the box to further improve the performance of the model.

We are inspired by the above ideas, the penalty item of box-related information is also added to achieve the purpose of improving the regression accuracy of the model. Two aspects should be taken into account when designing the penalty item. On the one hand, it is necessary to introduce appropriate box information to add the penalty item, such as center distance, aspect ratio, etc. To help the convergence of the IoU loss, the error information needs to be consistent with the IoU loss gradient. If the gradient direction of the penalty term is inconsistent with the original direction, the result may be counterproductiv. On the other hand, the appropriate form of the error information in the penalty term is selected. The form should have scale invariance to avoid adverse effects such as gradient explosion and gradient disappearance during model training. Otherwise, it may reduce the generalization ability of the model and maximize the effectiveness of the error information.

Based on the above considerations, the box information selected is the diagonal distance between the two boxes. As shown in Figure 6. Where *r* is the diagonal of the rectangle intersecting the two boxes, and *c* is the diagonal of the smallest bounding rectangle of the two boxes.

Since the distance information is sensitive to scale, it is necessary to normalize the distance information to make the penalty item scale-invariant. The expression can be shown as:(7)$$d_{1}=\frac{r^{2}}{c^{2}}$$

However, the ratio increases with the box intersection ratio. The gradient direction is opposite to the one of CIoU, so it is necessary to take the opposite gradient direction. Since the ratio of this form has a range of [0, 1], the form of the basic penalty term should be:(8)$$\delta =1-\frac{r^{2}}{c^{2}}$$

#### 3.1.2. Final Penalty

After determining the basic form of the penalty item, an appropriate form is needed to improve the performance of CIoU. Since the basic form of the penalty term has already prossessed the properties of scale invariance and gradient in the same direction as CIoU, the final form of the penalty term only needs to consider smoothness.

For SmoothL1 function that gets the advantages of L1 and L2 functions in stages, when the error in the initial stage of training is large, a stable gradient can be maintained, avoiding problems such as gradient explosion or disappearance. In the process of gradual stability in the later stage of the model, the gradient gradually decreases with the error. The learning rate is reduced, so it is easier for the model to converge to the extreme point. Although Smooth L1 has many advantages, its own form does not allow it to exert its performance at all stages of training, and it can only exert the performance of the norm of L1 or L2 at a certain stage. To obtain the advantages of partial L1 and L2 norms simultaneously, a basic norm penalty term is proposed based on Smooth L1:(9)$$R=\gamma \left(\delta -0.5\times \beta \right)+0.5\left(1-\gamma \right)\delta^{2}/\beta$$

Among γ Is the weight of penalty terms based on the L1 and L2 norm forms, which is used to adjust the role of two penalty terms, β Is the error value when the model accuracy tends to stabilize. The function graph is shown in Figure 7.

The final form of the penalty term is mainly composed of two parts. The first part is a penalty term based on the L1 norm form, the latter part is a penalty term based on the L2 norm form. The combination of L1 and L2 functions can fully utilize their respective advantages. The sparsity of L1 functions can filter noise and outliers, while allowing faster parameter updates. The L2 function can pay more attention to the punishment of large errors, thereby providing stronger control over the “averaging” characteristics of the dataset, improving the generalization performance of the model, and making the model more robust.

The advantage of this form of penalty project is that the L1 and L2 norms can be obtained throughout the model training process. In addition, additional effective box information is added to the original regression function to improve the accuracy of bounding box regression. According to the different performance of different models in the early and late stages of the training process, two parts of the penalty term and a coefficient of 1 are given as weights, respectively. Adjusting the role of each part in model training can make the frame regression function adapt to multiple object scales, enable accurate positioning of object boxes of different sizes, and improve the accuracy of detection. The derivative expression of the penalty term is as follows:(10)$$\frac{dR(\delta )}{d_{\delta }}=\gamma +\frac{1-\gamma }{\beta }\delta$$

Figure 8 shows that the penalty term obtains different degrees of L1 and L2 norms through different values of γ. In view of the possibility of gradient explosion in the early stage of the model training, the value of γ should be increased to enhance the effect of the L1 form of the penalty term, while suppressing the effect of another part of the penalty term. For example, γ = 0.9, when the model error changes, there is only a tiny change in the gradient value. In the later stage of model training, the penalty term can still provide a relatively stable gradient value. The model can converge to a stable extreme point.

Since the penalty term is proposed based on two norms and contains some of their advantages, the improved bounding box regression loss function is called Norm CIoU(NCIoU):(11)$$\mathrm{NCIoU}=IoU+\frac{p^{2}\left(b,b^{gt}\right)}{c^{2}}+\alpha v+\lambda R\left(\delta \right)$$
where λ is the weight of the penalty term in the entire function. By adjusting γ, the penalty term obtains the advantages of L1 and L2 to varying degrees. Using L1 and L2 loss functions simultaneously can change both the penalty intensity of the model and the penalty method of the model. This makes the model more flexible and can better adapt to different data distributions and actual situations. Its main contribution is to introduce additional box information into the regression function, helping more prediction boxes continuously approach the real box, improving the accuracy of box regression, thereby achieving the goal of improving model accuracy. Experiments have shown that multiple evaluation indicators of the loss functions such as CIoU and DIoU that add this penalty term have been improved.

### 3.2. Improvement of Confidence Loss Function

In practical situations where only a small portion of an image has an object. The confidence loss function of YOLOv4 adds weights to the loss without an object to reduce the contribution of this loss value, which may otherwise cause the network to tend to predict cells without an object, thus affecting the model accuracy.

Using the BCE function as a confidence loss function suffers from the problem of biasing the learned algorithm to identify no object. Since most of the image content is background, most of the samples in the images are negative samples, which will amplify the loss of negative samples and cause the training results to be biased towards negative samples. Although weight is given to limit this, it still has an impact on the final result. Similarly, if the training uses a dataset with much more object than background information in the images, the algorithm will appear biased in the actual detection process to identify the background information as the object to be detected.

When calculating this part of the loss, MSE is calculated as [0, 1] and the model is not biased toward the presence or absence of the object. Experiments with the YOLOv4 of algorithms show that MSE does not perform worse than the BCE function. Combining MSE and BCE as confidence loss functions to calculate both parts of the confidence loss in a targeted manner has better performance than using BCE or MSE alone. The combined confidence loss function is shown below:(12)$$\begin{array}{c} {\mathrm{loss}}_{\mathrm{conf}\;}=\sum_{i=0}^{S^{2}}\sum_{j=0}^{B}I_{ij}^{\mathrm{obj}\;}\left(C_{i}^{j}\log\left({\hat{C}}_{i}^{j}\right)+\left(1-C_{i}^{j}\right)\log\left(1-{\hat{C}}_{i}^{j}\right)\right)+\\ \lambda_{\mathrm{noobj}\;}I_{ij}^{\mathrm{noobj}\;}{\left(C_{i}^{j}-{\hat{C}}_{i}^{j}\right)}^{2} \end{array}$$

BCE is primarily used to handle confidence loss with objects, while MSE is used to handle confidence loss without objects. The combination of the two loss functions can improve the balance between the model’s detection accuracy and the false alarm rate for objects, further improving the model’s performance. Specifically, when a prediction box contains a target object, the BCE loss function is used to measure the difference between the confidence of the prediction box and the real box. By optimising the binary loss of the prediction box, it can help the model to better identify the target object.

As the object-free scenes are more trivial and complex than the object scenes, optimization using only BCE can easily lead to problems such as false recognition detection of the model. The introduction of the MSE loss function, on the other hand, can help the model to better handle the prediction box in targetless scenes, reduce the response of the model to existing noise and other disturbances, and reduce the false detection rate of the model. Therefore. Using both BCE and MSE as confidence loss functions in the object detection model can better handle diverse object detection scenarios and improve the robustness and accuracy of the model.

### 3.3. Feture Fusion Network

Although the feature pyramid improves the use of shallow information by fusing different levels of feature maps, it loses some object information during the convolution and pooling processes of fusion. In particular, the following sampling operations, which produce low-resolution feature maps, lose most of the fine feature information. To alleviate this problem, we added cross stage lateral connections to the feature pyramid inspired by residual networks. It will make up for some of the missing object information. In addition, a lightweight hybrid attention HAMNet was added to the feature pyramid lateral fusion path to dig deeper into the feature information of feature maps at different scales. This highlights the object information in order to retain more of it during convolution and pooling, etc. Applying these two methods on the basis of the feature pyramid obtained a novel feature pyramid network based on the attention module(AM-NFPN). Its framework diagram and details are shown in the neck section in Figure 9.

The cross stage horizontal connection path fuses the feature map not fused at the same level with the fused feature map, which can enrich the location information and semantic information of the feature map at the same level, and further strengthen the use of information. More importantly, cross stage connections provide a new path. If the feature fusion effect of the original path is not ideal, the unprocessed feature map detection effect of the new path should be better than the original network. After the two paths are fused, the feature information of the final feature map should not be worse than the original network, the final detection effect should not be worse than the original network.

There are three cross stage lateral join paths, each of which is fed with three feature maps that have not been fused before entering the neck network. No operations are performed during their lateral join to ensure that more information about the object is fused without adding additional computational effort and thus without increasing the learning time of the model. The unfused input features are Concat fused with the output of the feature pyramid at the same layer, and then output through a 2D convolution block, which starts with a 2D convolution with a kernel of 1 and a step size of 1 for the purpose of integrating the channels, followed by batch normalization for normalizing the feature map and equalising the non-linear features at each layer, finally, a LeakyReLU activation function to enhance the representation of the non-linear features of the feature map. AM-NFPN enhances the performance of the model while adding only a small number of parameters to the model that do not affect the balance between model accuracy and precision.

In AM-NFPN, HAMNet is used in conjunction with SPP [38] to reduce noise information from upsampling process for the feature map $P_{5}$ output from the backbone network. For feature maps $P_{3}$ and $P_{4}$ of the backbone network output, more object information is retained in the downsampling process using HAMNet. The purpose of cross stage connectivity is to compensate for some of the lost object information, and the use of HAMNet aims to reduce the loss of target information during the fusion of multi-scale feature maps.

Using function $f_{1}$ to represent CBL and Concat operations and function $f_{2}$ to represent Concat and HCH operations. At the same time, the CBL representing convolution, normalization and LeakyRule activation function is represented by conv2D. Then $f_{1}$ and $f_{2}$ can be represented by the following equation:(13)$$\begin{array}{c} f_{1}=Conv2D\left(Concat\left(X_{1},X_{2}\right)\right) \end{array}$$
(14)$$\begin{array}{c} f_{2}=HAM\left(Conv2D\left(HAM\left(Concat\left(X_{1},X_{2}\right)\right)\right)\right) \end{array}$$

The weights of the two parts of the feature map are $\omega_{x,1}$ and $\omega_{x,2}$, respectively, when connected horizontally. The three output feature maps for the AM-NFPN network can be expressed as:(15)$$\begin{array}{c} P_{3}^{\mathrm{out}\;}=f_{1}\left(\omega_{3,1}\cdot f_{2}\left(\mathrm{Conv}2D\left(P_{3}\right),P_{4}^{up}\right),\omega_{3,2}\cdot P_{3}\right) \end{array}$$
(16)$$\begin{array}{c} P_{4}^{\mathrm{out}\;}=f_{1}\left(\omega_{4,1}\cdot f_{1}\left(\mathrm{Conv}2D\left(P_{3}^{\mathrm{down}\;}\right),f_{2}\left(\mathrm{Conv}2D\left(P4\right),P_{5}^{up}\right)\right),\omega_{4,2}\cdot P_{4}\right) \end{array}$$
(17)$$\begin{array}{c} P_{5}^{\mathrm{out}\;}=f_{1}\left(\omega_{5,1}\cdot f_{1}\left(P_{4}^{\mathrm{down}\;},\mathrm{Conv}2D\left(HSH\left(P_{5}\right)\right)\right),\omega_{5,2}\cdot P_{5}\right) \end{array}$$

## 4. Experimental Results and Analysis

To evaluate our method, we conduct experiments on the RFM. To ensure the comparability of the experimental results, all the experimental training set and test set pictures in this article are exactly the same. The experimental environment is always the same. The experimental equipment is shown in Table 1.

The experimental dataset has an input image resolution of 416 × 416, a hyperparameter bach size of 4, an initial learning rate of 0.01, an iteration round epoch of 50 and a SGD optimizer for gradient descent.

### 4.1. Date Set and Evaluation Index

The RFM dataset contains a total of 12,133 images, of which 80% are used as the training and validation set and the remaining 2427 images are used for testing. In total, 8738 images are used for training and 968 images for validation. The dataset is divided into three categories: the face category, the normative mask category and the irregular mask-wearing category. The data images are not entirely ideal. To better match the actual human activity, some images are not masked but have their faces obscured, so these object are labelled as the face category. The mask dataset was created with a face category number of 0, labelled face; an irregularly worn mask number of 1, labelled WMI; and an irregularly worn mask number of 2, labelled face_mask.

To obtain more accurate detection results, reasonable prior bounding box were obtained for the multi-scale objects by clustering, as shown in Table 2 for the RFM dataset.

The following indicators are used to evaluate the performance improvement of the proposed method on YOLOv4. Mean Average Precision (mAP), which is the average of the accuracy of all categories, the expression is:(18)$$mAP=\frac{1}{N}\sum_{i=1}^{N}AP\left(i\right)$$
where *N* represents the number of all categories, AP is the average accuracy rate of a category and the expression is:(19)$$AP=\int_{0}^{1}P\left(R\right)dR$$
where *P* (Precision) is the accuracy rate of a certain type of sample, *R* (Recall) is the recall rate, and their expression are:(20)$$P=\frac{TP}{TP+FP}$$
(21)$$R=\frac{TP}{TP+FN}$$
where TP represents the number of positive samples that are correctly divided; FP is the number of positive samples predicted but actually negative samples; FN refers to the number of negative samples predicted but actually positive samples.

F1-Score is a measure of classification problems and is often used as the final indicator for multi-classification problems. It is the harmonic mean of Precision and Recall.
(22)$$F_{k}=2\frac{\;\mathrm{recall}{\;}_{k}*\;\mathrm{Precision}{\;}_{k}}{\;\mathrm{recall}{\;}_{k}+\;\mathrm{precision}{\;}_{k}}$$

The Log-Average Miss Rate (LAMR) indicates the omission of the test set in the dataset. Larger LAMR, more missed object, smaller LAMR, and less missed object, also indicate better model performance.The miss rate expression is:(23)$$\mathrm{MR}=\frac{FN}{FN+TP}$$

### 4.2. Ablation Studies

A combination experiment of different modules is set up in order to analyze the influence of the NCIoU loss function, the improved confidence function and AM-NFPN on the model performance. Results of the experiment are shown in Table 3.

The first group in the ablation study table is the benchmark model experimental data. Compared to the benchmark model, when using the three strategies alone, all evaluation indicators have improved. The NCIoU strategy has the best improvement effect. Specifically, mAP@.5:.95 and AP75 increased by 3.34% and 4.91%, respectively. The least effective improvement was achieved by using the confidence scheme, but this resulted in at least a 2% improvement in all indicators compared to the benchmark model. All three sets of experiments using a combination of the two strategies showed better results than one. In all three sets of experiments, the model with the NCIoU strategy was better than the model without it. It can be seen that the NCIoU strategy has the greatest impact on the model and the best improvement effect among the three strategies. When the three strategies are used together, the comprehensive performance of the model is the best, with optimal performance across multiple indicators, mAP@.5:.95, AP75 and Recall have been improved by 3.98%, 5.69% and 5.08%, respectively, and the missed detection rate has been reduced by 5.7%. It is the final algorithm used to detect faces and identify whether masks are worn in a standardized manner.

Figure 10 shows test images with pixel values from 64 to 960, and applied CIoU and NCIoU functions on the image pyramid. The object in the test image is a face. The results show that the lower the pixel value, the more difficult the loss function is to process the object. As discussed in Section 3.1, rich bounding box information can provide more accurate prediction boxes to help detect more objects, accompanied by higher confidence. In addition, additional box information does not only make the prediction box closer to the real box, but also adjust the aspect ratio of the bounding box to obtain a higher intersection ratio to make the detection effect more accurate, which is fully demonstrated in the test diagram. It is worth noting that NCIoU can still provide a more reasonable prediction box compared to CIoU in extreme situations.

The advantage of NCIoU is that the parameters can be adjusted to adapt the function to changing detection scenarios, and varying α allow the function to obtain different characteristics to cope with complex scenarios. The variation of each index of NCIoU for different values of α is shown in Figure 11. There are five groups of experiments with α values of 0.1, 0.3, 0.5, 0.7, and 0.9. It can be seen from the figure that each metric has good improvement when α is 0.3 and 0.7, but the F1 score decreases abruptly when α is 0.7. Overall, the model performance is optimal when α is 0.3. When α is 0.5, the model has the least improvement effect, but the performance of NICIoU is also better than CIoU at this time.

CIoU is gaining a high degree of recognition, and many better functions have been proposed [39,40,41] based on it since then. We conducted some comparative experiments on RFM with some bounding box regression functions, as shown in Table 4. EIoU is the most effective of these published methods, but has a 2% difference in several metrics compared to NCIoU.

The penalty term applied to CIoU is also applied to the commonly used bounding box regression loss functions DIoU and GIoU to verify the effect of the penalty term on the other functions, as shown in Table 5.

It can be seen that all the metrics of both loss functions are improved after adding the penalty term, with the most pronounced performance on mAP@.5:.95. Overall, the penalty term improves GIoU in a away that is better than DIoU, which is due to the fact that GIoU contains less bounding box information than DIoU itself, so it plays a bigger role when adding box information, and also shows that the more information the loss function contains, the better the performance. The α parameter of the penalty term can be adjusted for different functions to obtain better results for each function, and the generalization performance of the function can be improved by adjusting the parameter λ for different datasets to make the function more adaptable to different scenarios.

As shown in Figure 12, the actual detection map is visualized with the heat map; the blue boxes represent irregular mask wear and the green boxes represent regulated mask wear. The heat map shows that the baseline network does not pay any attention to the smaller objects, and less information makes the model think that the probability of the presence of the object here is low and the missed detection is more prominent.

After AM-NFPN compensates and highlights the object information, the feature map can obtain enough information for even smaller objects to make the model identify the presence of an object and thus improve the detection accuracy. In addition, since there are common attributes and features for the three categories of face, normative mask wear and irregular mask wear, a mask can be used to describe whether or not the mask is normatively worn in both cases when the mask wear is between or close to the defining line of the two categories, it is easy to cause the model to return two different categories of prediction boxes for the same object. This situation is especially prominent when the information is not sufficient. The benchmark module lacks sufficient detailed information to predict the irregular category as both normative and irregular mask wear, while the ability to incorporate more target information AM-NFPN greatly avoids this situation and reduces the false detection rate.

Commonly used feature fusion networks are to fuse multi-scale feature maps in a ratio, such as FSSD’s [42] feature fusion module(FFM), FPN and PANet. We compared these methods experimentally with the Pyramidal feature hierarchy(PFH) method on RFM, as shown in Table 6. It can be clearly seen that AM-NFPN achieves better results. This better performance mainly lies in the loss of less object information during the fusion process, which is mainly shown in the actual detection results for small objects, blurred objects and obscured objects in the gap.

In addition, several confidence loss functions commonly used today are also compared for experiments on RFM, as shown in Table 7. The Focal function with the worst overall performance obtained better results on Precision, which is the result of sacrificing the performance of Recall. The difference between the overall performance of MSE and BCE is very small, and it is difficult to have a significant difference between the performance of the two functions in practical applications. The combination of these two functions with targeted calculation of confidence loss can realize the performance of the two functions to some extent and show more impressive results.

### 4.3. Comparative Experiment

As shown in Table 8, the experimental comparisons of several models with different sizes of input images are presented. It can be seen that the larger the input image is for each model, the better the training results are. Several evaluation metrics can get some improvement, which is due to the higher resolution of the input image, and the image packs more detailed information, but a larger image will take more time to train.

Our algorithm achieves the highest values for AP75 and mAP@.5:.95; this is still 2% higher than the highly acclaimed YOLOv7-X, which achieved the second best result in the comparison experiments for three sizes of input images. Compared with the benchmark model YOLOv4, there are good improvements in several metrics, and the improvement in model performance is mainly due to the inclusion of more detailed information in the final feature map to be detected. For the YOLOX algorithm with improved CSPDarkNet-53 network, although its S, M, and L type feature extraction networks are relatively small, the accuracy rate is insufficient in detecting faces and masks, and the mAP is greatly reduced under different thresholds. For the most stringent mAP@.5:.95, when the input image size is 416 × 416, our algorithm is 2.55% higher than the second best YOLOv7-X, 12.64% higher than the worst performing Faster R-CNN, and still 12.23% higher compared to the SSD with 512 × 512 input image.

We selected YOLOv4 and YOLOv7-X, which have better algorithm performance, and did the visualization and analysis of the detection results of each algorithm in the same experimental environment, as shown in Figure 13. The standard mask object is marked as face_mask in green, followed by the number of confidence degrees; the irregular mask-wearing object is marked as WMI in blue; the face object is marked as face in red.

Compared with the other two algorithms, the detection results of our algorithm are better, not only being able to detect objects missed by the other two algorithms, but also objects detected by all algorithms tend to have a higher confidence level. YOLOv4 and YOLOv7-X are able to detect most of the objects, but due to the influence of factors such as object attitude, focal distance during imaging and occlusion, there are cases of missed detection, and these factors are precisely the key issues that limit the accuracy of object detection algorithms. While our algorithm is more capable of coping with unfavorable factors and is able to detect small or occluded objects that are difficult to detect by other algorithms, reducing the overall misdetection rate and false detection rate, and is able to adapt to more and more complex scenarios.

The heat map visualization of this paper with YOLOv4 and YOLOv7-X is shown in Figure 14, where the darker the color, the higher the confidence level of the object. It can be seen that the heat map of this algorithm has the darkest color in the region with objects, and has a higher confidence level when detecting the objects, which is confirmed by the detection results in Figure 13. Compared with the other two algorithms, our algorithm focuses on more objects and has a certain degree of attention to the objects missed by other algorithms, thus detecting more objects with higher detection accuracy. Although the color of YOLOv4 heat map is lighter than ours, it is darker than YOLOv7-X, and the focus range is smaller, which means it has higher confidence but the border position is not accurate enough.

Measuring the superiority of an algorithm mainly requires not only considering the detection accuracy of the algorithm, but also weighing the detection speed. For detecting faces and masks in public places, an algorithm with excellent accuracy and speed is needed; high accuracy and low speed or vice versa will not accomplish the desired task. For this purpose, the accuracy and speed of several algorithms are shown in Figure 15. It shows that the detection accuracy of our algorithm is the most competitive. Although the detection speed is not the fastest, it is only less than two milliseconds compared to the YOLOv4 algorithm, which has achieved a high balance of accuracy and speed and is widely used, and slightly better than YOLOv7-X in terms of speed and accuracy. The YOLOX algorithm increases the detection speed as it reduces the complexity of its backbone feature extraction network, but the accuracy rate decreases and it is difficult to complete the detection task. YOLOX-X algorithm has good speed and accuracy, but it still has some gap from our algorithm. Other algorithms such as SSD and EfficientDet can meet the requirements in terms of detection speed but still have a certain gap in accuracy. Overall, our algorithm is a balanced algorithm in terms of detection accuracy and degree and can meet the requirement of detecting faces and masks in real time.

The Figure 16 shows the comparison of our algorithm and YOLOv4 algorithm detection results under different occlusion level, illumination and distance. For varying degrees of facial occlusion, YOLOv4 again appears to return two detection boxes, while the false detection is more serious. Our algorithm detects occlusions better than YOLOv4, but more difficult facial occlusion cases also show examples of false detection. YOLOv7-X does not appear to show two predicted boxes, but instead does not detect objects, as shown in the two images with more occlusions. Our algorithm performs consistently with the YOLOv4 algorithm at different distances and illumination, and the given examples all detect the object correctly. For other examples such as less illumination or longer distances, all algorithms fail to detect the object. The farthest distance image and the darkest image given in the figure YOLOv7-X both fail to detect the object. Our algorithm outperforms the other algorithms in the comparison experiments, and the combined detection performance of our algorithm is better than the other algorithms for different scenarios, which is beneficial for more applications.

## 5. Conclusions

In view of some of the shortcomings of the currently widely used YOLOv4 algorithm, we propose a new bounding box regression function NCIoU. Furthermore, BCE and MSE functions are used to calculate the confidence loss in a targeted manner. The more efficient feature fusion network AM-NFPN is able to obtain feature maps that contain more information. It is worth noting that we create a dataset containing 12,133 images. It contains many pictures of real complex scenes. The results show that the performance of our model has been improved in many ways compared to the original model without reducing the speed, and the Recall, AP75 and mAP@.5:.95 have been improved by 5.08%, 5.69% and 3.96%, which means that it can be better in practical applications. The detection effect can be applied to more fields.

For the automated intelligent detection of faces and mask-wearing high-precision and high-efficiency needs, enhancing the feature extraction capability of the algorithm is the approach to meet the requirements of high accuracy. This general approach of deepening the network will bring about a reduction in the detection speed, lightweight and high-precision detection algorithms and can be the direction of further research.

## Figures and Tables

**Figure 1 sensors-23-04612-f001:**
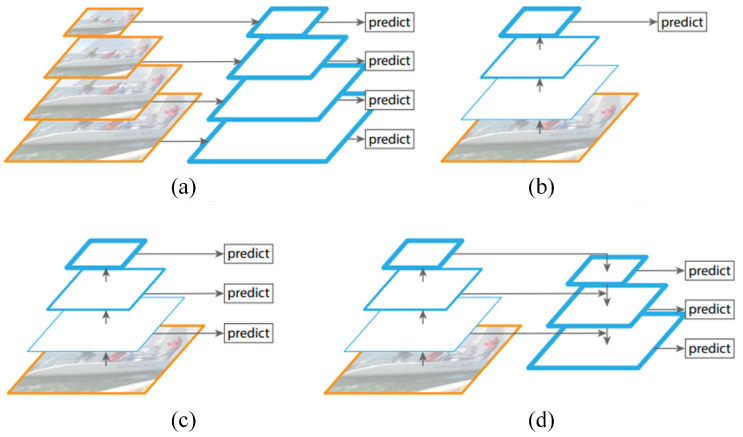
Commonly used feature integration approaches [23]. (**a**) Featurized image pyramid. (**b**) Single feature map. (**c**) Pyramidal feature hierarchy. (**d**) Feature pyramid network.

**Figure 2 sensors-23-04612-f002:**
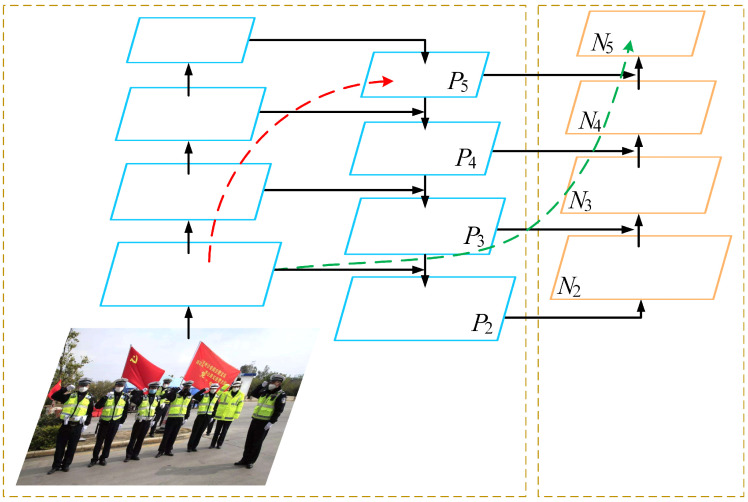
Path aggregation network.

**Figure 3 sensors-23-04612-f003:**
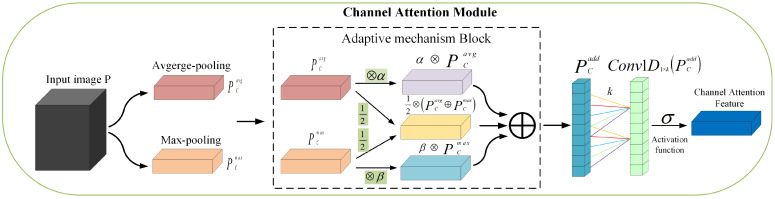
The channel attention sub-module of HAMNet.

**Figure 4 sensors-23-04612-f004:**
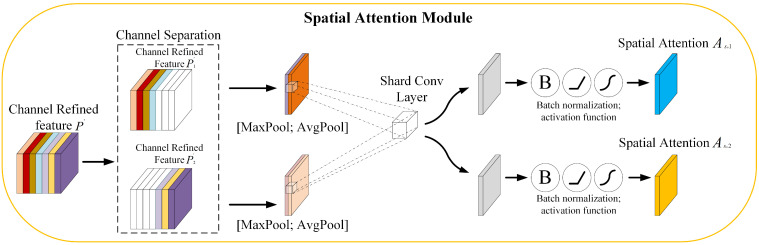
The spatial attention sub-module of HAMNet.

**Figure 5 sensors-23-04612-f005:**
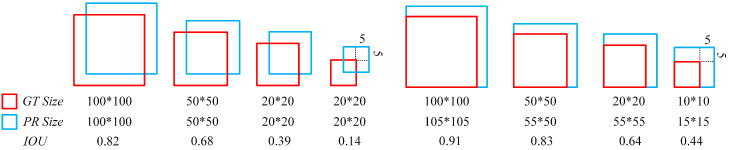
IoU between the bounding box and the prediction box under the given error.

**Figure 6 sensors-23-04612-f006:**
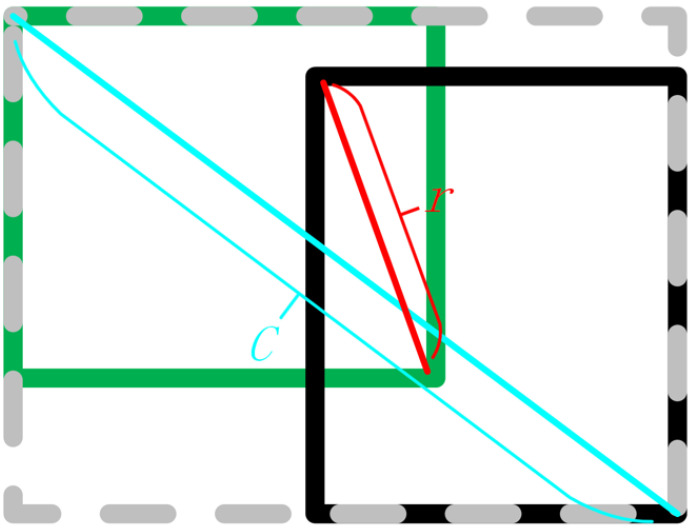
Selected box informatiom.

**Figure 7 sensors-23-04612-f007:**
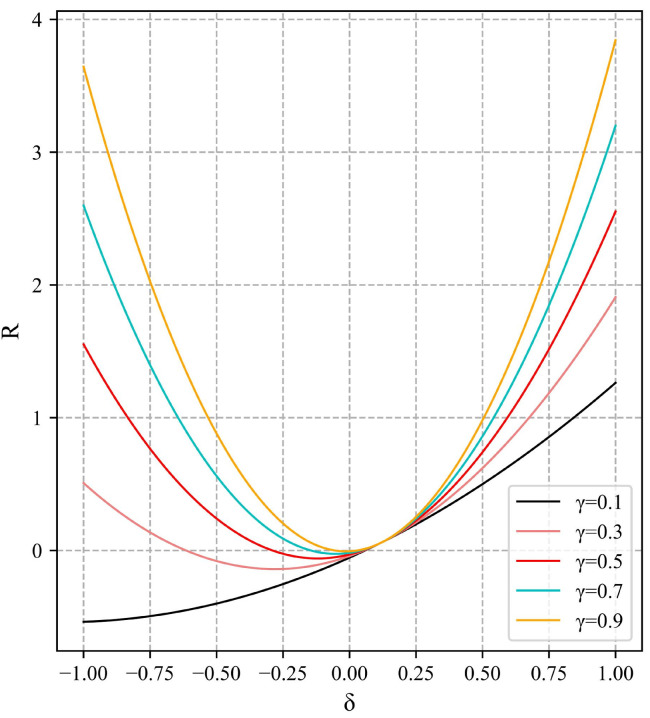
Penalty function graph.

**Figure 8 sensors-23-04612-f008:**
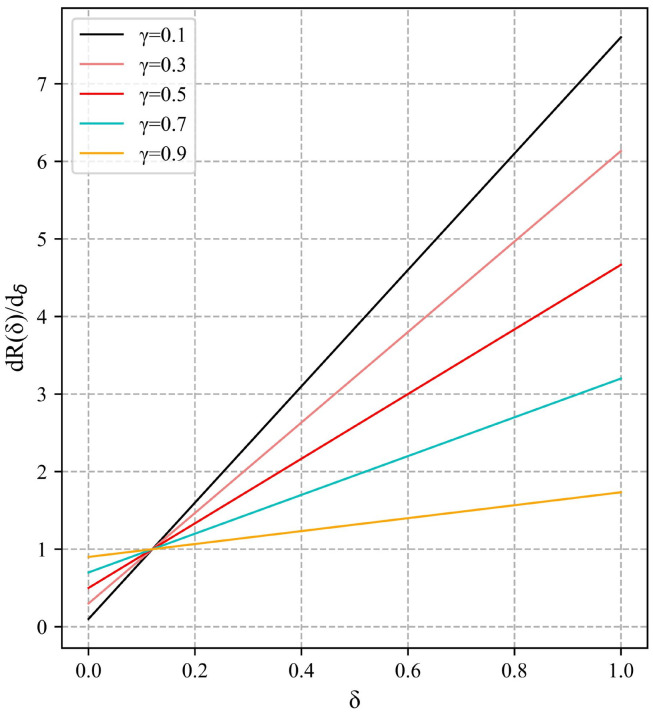
Penalty derivative graph.

**Figure 9 sensors-23-04612-f009:**
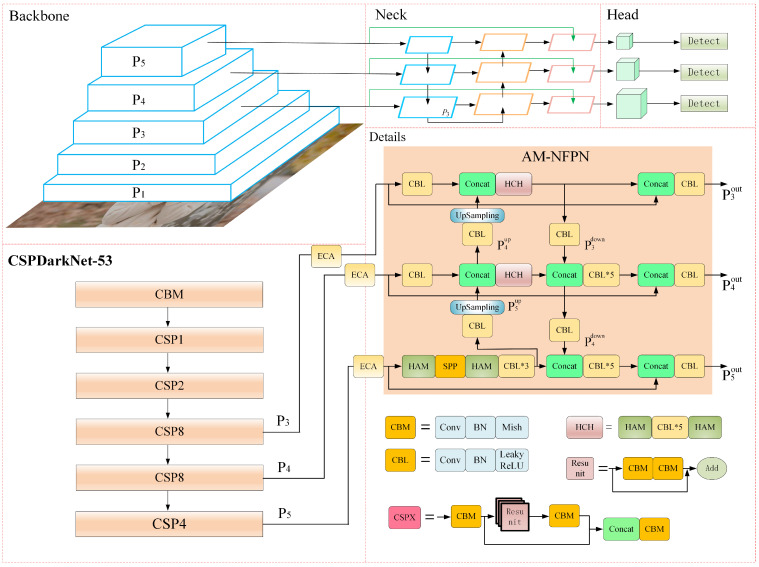
Overall framework diagram of the model.

**Figure 10 sensors-23-04612-f010:**
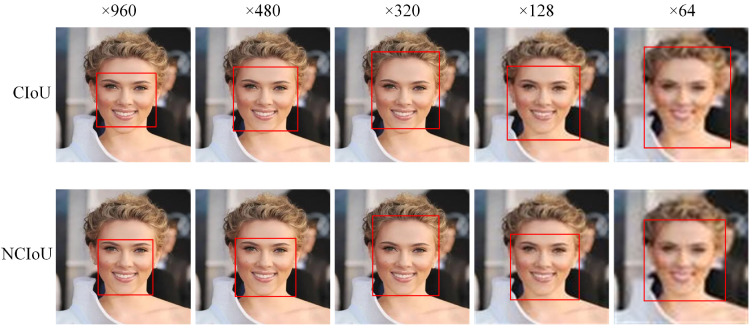
Detection performance of NCIoU and CIoU under different pixel sizes.

**Figure 11 sensors-23-04612-f011:**
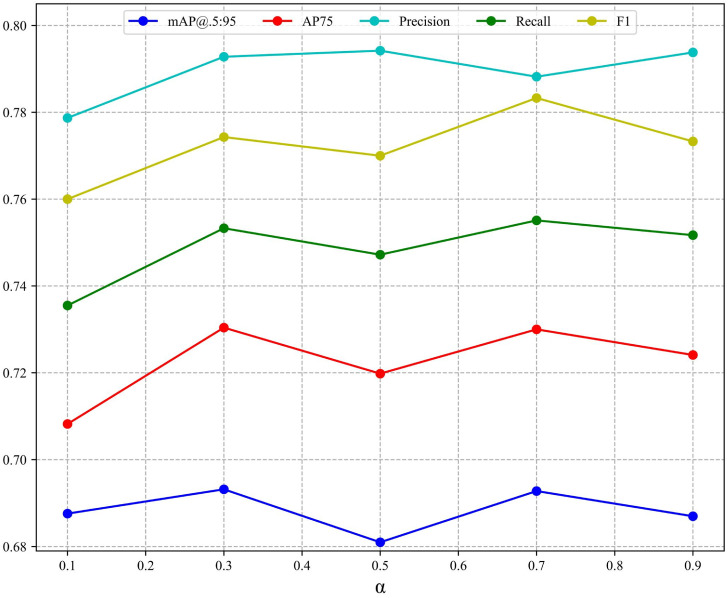
Different α Changes in various indicators at value.

**Figure 12 sensors-23-04612-f012:**
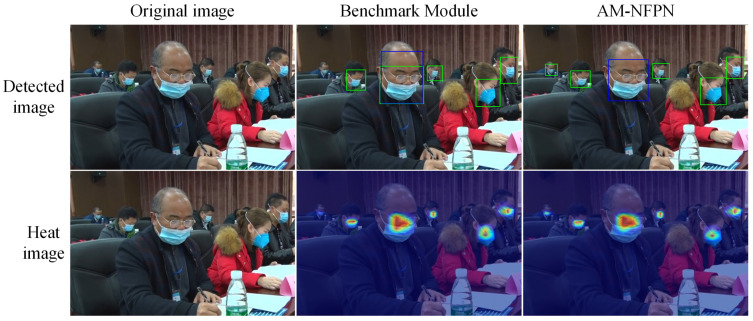
AM-NFPN and benchmark module detection diagram and thermal diagram visualization.

**Figure 13 sensors-23-04612-f013:**
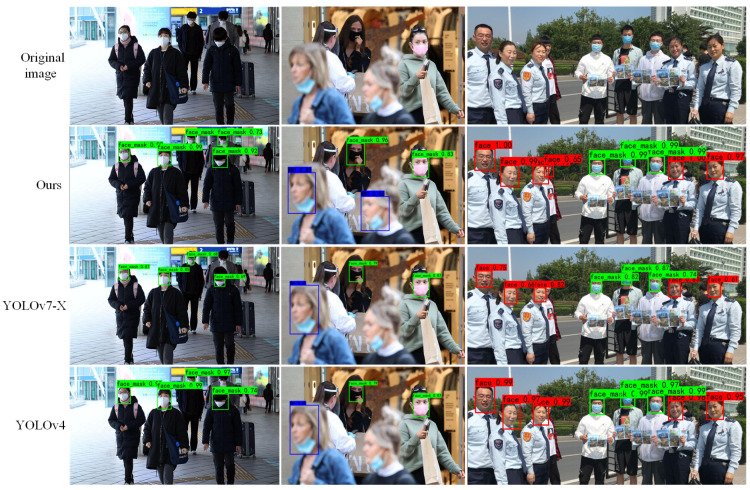
Visualization of detection results for different models.

**Figure 14 sensors-23-04612-f014:**
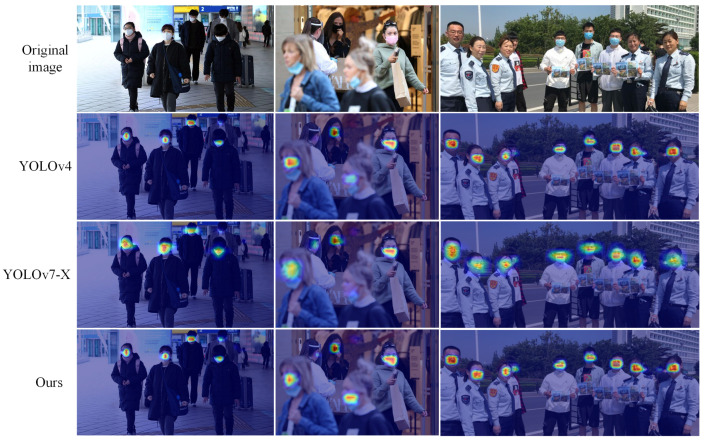
Visualization of heatmaps for different models.

**Figure 15 sensors-23-04612-f015:**
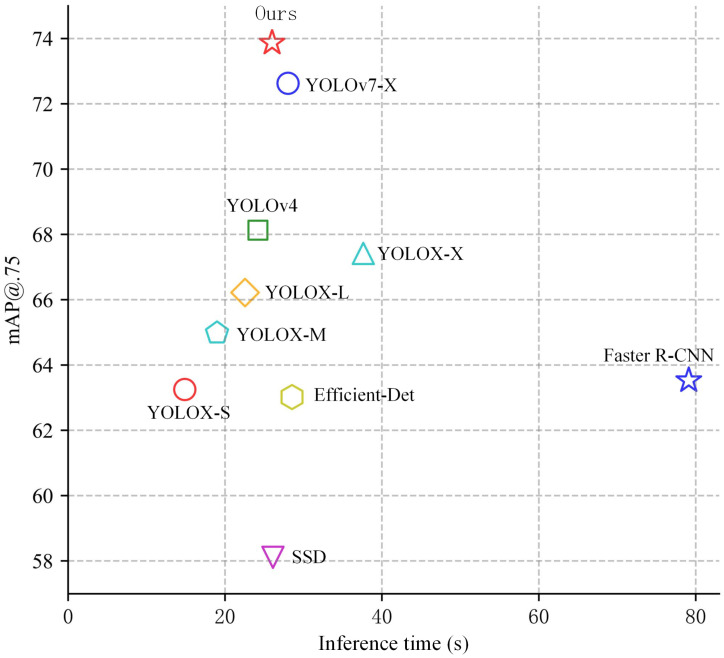
Comparison of detection accuracy and speed among multiple algorithms.

**Figure 16 sensors-23-04612-f016:**
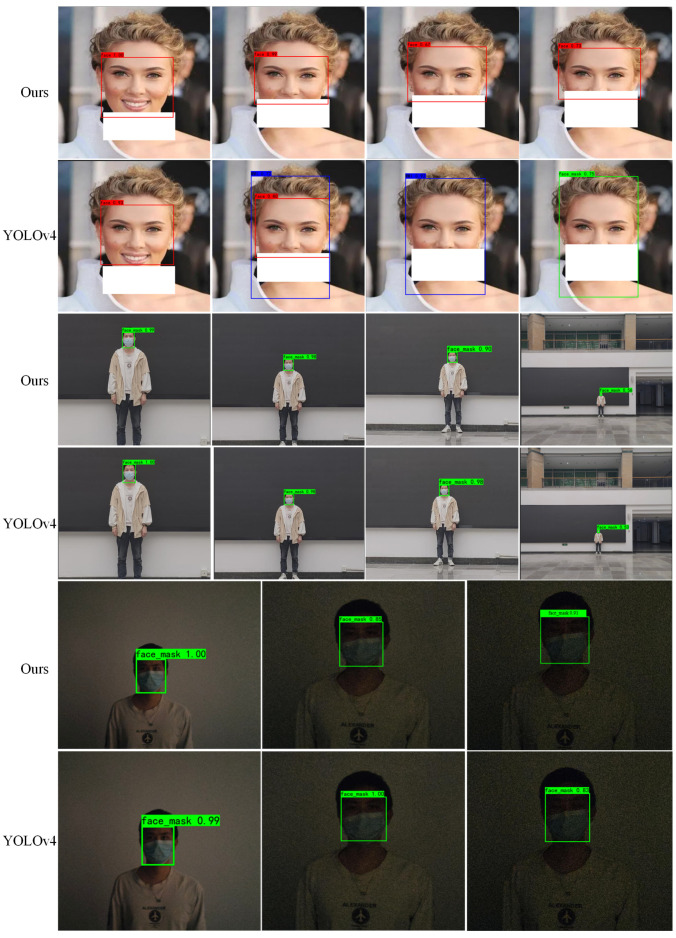
Detection results with different occluded faces, light and distance.

**Table 1 sensors-23-04612-t001:** Experimental environment.

Name	Configuration
Operating system	Window 10
CPU	Intel(R)i7-11700K
GPU	RTX3060, 12 GB
frame	Pytorch, Keras, Tensorflow
Programming	Python 3.7
Dependent package	CUDA 11.1, CUDNN 8.0.4

**Table 2 sensors-23-04612-t002:** Prior bounding boxes of feature maps with different scales.

Feature Map Size	Receptive Field	Prior Bounding Box
		9 × 15
13 × 13	Samll object	16 × 25
		24 × 41
		42 × 63
26 × 26	Middle object	81 × 107
		162 × 182
		218 × 237
52 × 52	Large object	223 × 213
		240 × 245

**Table 3 sensors-23-04612-t003:** Network performance(%) of different module. The checkmark represent the use of this method.

NCIoU	Confidence	AM-NFPN	mAP@.5:.95	AP75	Recall	Precison	Larm	F1
			65.98	68.13	71.59	75.39	38.7	74
✓			69.32	73.04	75.33	78.28	34.3	77
	✓		68.08	72.45	75.40	76.49	35.3	76
		✓	68.13	72.28	74.99	78.72	34.7	76
✓	✓		68.85	73.27	75.01	78.41	34.3	77
✓		✓	68.20	71.70	74.6	78.19	34.7	77
	✓	✓	66.58	72.64	75.60	76.45	35.7	76
✓	✓	✓	69.96	73.82	76.67	78.02	33.0	77

**Table 4 sensors-23-04612-t004:** Comparison(%) of bounding box regression functions on RFM.

Function	mAP@.5:.95	AP75	Recall	Precision	Larm	F1
CIoU	65.98	68.13	71.59	75.39	38.70	74
ICIoU	65.77	70.64	73.78	77.19	36.67	75
EIoU	67.22	71.89	74.29	79.06	35.02	77
SIoU	64.16	69.43	72.43	76.60	37.33	74
NCIoU	69.96	73.82	76.67	78.02	33.01	77

**Table 5 sensors-23-04612-t005:** Performance comparison(%) of GIoU and DIoU after adding penalty terms.

Function	Penalty	mAP@.5:.95	AP75	Recall	Precision	Larm	F1
DIoU		66.15	69.08	72.03	74.93	37.7	73
DIoU	✓	68.79	70.81	73.49	75.93	36.3	74
GIoU		65.32	67.10	70.52	72.43	38.9	72
GIoU	✓	67.28	69.02	71.94	73.82	37.3	73

**Table 6 sensors-23-04612-t006:** Multiple feature fusion network comparison(%) experiments on RFM.

Function	mAP@.5:.95	AP75	Recall	Precision	Larm	F1
FFM	64.52	68.33	73.96	71.34	38.01	72
FPN	64.37	66.83	74.09	69.07	39.33	72
PFH	60.08	63.08	66.84	71.47	43.30	69
PANet	65.98	68.13	71.59	75.39	38.70	74
AM-NFPN	69.96	73.82	76.67	78.02	33.01	77

**Table 7 sensors-23-04612-t007:** Experimental comparison(%) of multiple confidence functions on RFM.

Function	mAP@.5:.95	AP75	Recall	Precision	Larm	F1
MSE	65.55	68.58	72.32	75.79	39.67	58
Focal	64.76	64.56	55.51	81.59	46.33	66
BCE	65.98	68.13	71.59	75.39	38.70	74
Ours	69.96	73.82	76.67	78.02	33.01	77

**Table 8 sensors-23-04612-t008:** Performance comparison(%) of multiple model on RFM.

Module	Backbone	Size	mAP@.5:.95	AP75	Recall	Larm	F1
YOLOv4	CSPDarkNet-53	608 × 608	67.24	70.40	73.07	36.3	74
		416 × 416	65.98	68.13	71.59	38.7	74
		320 × 320	63.99	66.39	69.33	39.7	71
YOLOv7-X	-	608 × 608	69.12	74.03	76.86	32.2	78
		416 × 416	67.41	72.63	75.16	33.3	78
		320 × 320	66.92	71.32	73.57	34.0	76
SSD	VGG-16	512 × 512	57.73	63.41	63.39	41.7	70
		300 × 300	53.26	58.11	58.63	46.3	65
Faster R-CNN	ResNet-50	608 × 608	59.34	65.83	67.91	38.3	66
		416 × 416	57.32	63.53	66.27	41.0	63
		320 × 320	49.45	54.46	61.05	54.0	58
EfficientDet	EfficientNet	768 × 768	60.68	66.63	66.77	39.3	73
		640 × 640	58.16	64.21	65.03	41.7	71
		512 × 512	55.85	63.02	64.77	42.7	70
YOLOX-X	Modified CSPv5	416 × 416	60.96	67.42	70.01	38.0	74
YOLOX-L		416 × 416	59.04	66.22	69.07	39.7	73
YOLOX-M		416 × 416	57.90	64.98	67.71	40.0	73
YOLOX-S		416 × 416	54.40	63.25	66.06	41.6	72
Ours	CSPDarkNet-53	608 × 608	71.02	75.57	77.79	32.0	78
		416 × 416	69.96	73.82	76.67	33.0	77
		320 × 320	67.56	72.59	75.41	35.3	76

## Data Availability

The data presented in this study are available on request from the corresponding author.

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
