# Peer review of "A Face Detection and Standardized Mask-Wearing Recognition Algorithm"

_sensors, 2023, doi:10.3390/s23104612_

Round 1

Reviewer 1 Report

In this paper, the authors proposed a new feature pyramidal network based on the attention mechanism to reduce the loss of object information that will be caused by sampling and pooling in convolutional neural networks. Based on the CIoU, a penalty function based on the norm is proposed to improve positioning accuracy, the new bounding box regression function is called NCIoU. Combination of two functions to calculate the confidence loss to mitigate the problem of the algorithm bias towards determinating no objects in the image. Moreover, they provided a data set for recognizing faces and masks(RFM) that includes 12133 realistic images. The data set contains three categories: face, standardized mask and non-standardized mask. Experiments conducted on the data set demonstrate that the proposed approach achieves [email protected]:.95 69.70% and 14 AP75 73.80%, outperforming the compared methods.  The interesting points can be found, however, the presented manuscript is not prrpared well. If the manuscript can be considered to be published, some revision have to be done. The comments are following:

1.Please polish the abstract. Please check the logic of abstract. Please add sentences to explain the meaning, the main points, the improvement and the promising application of the study. Plenty of detail data have given, however, in abstract, important procedures and results should be mentioned in simple manner. Please focus on the main points and the improvement of the study.

2.Please highlight the advance of the study in Introduction. Please explain the development and creative work. The literature review should be carefully considered.

3.English in this paper needs improvement, which can make this paper more like a journal paper. 

4.Figures 3-4 are not clear, which hardly can be read. Please replote them.

5.How can we obtian Eq.(5), and the basic theory and derivation need to be provided.

6.Compared with the other reports, the advantage should be given.

7.Compared with the published reports, the presented method should be given in the improvement methods of computing technology should be provided in a formulaic manner.

8.The virus detection technology is the key to this article, and how to do virus detection technology is the key. The development and application ofvirus detection technology can also be considered in this article, such as the application in detection.

*Opt. Lett. 47, 6065-6068 (2022) https://opg.optica.org/ol/abstract.cfm?URI=ol-47-23-6065

The details can be seen in the comments to the authors.

Reviewer 2 Report

The following aspects must be elaborated in more details:

-section Related work contains theoretical information - similar methods (with results must be described)

-section 2 contains more theoretical information - a summary of these information must be specified and only how these info are used / updated in the proposed methods

-experiments must be described in more details: what about occluded faced, different illumination conditions, distance from the camera - these aspects must be considered in provided results - discuss the results considering these aspects

-ablation study must be extended - it is better to use other existing results in order to compare the performance of the proposed method (and not to use only some existing other models)

Other aspects that must be updated:

-2.2. bounding box regression -> Bounding box regression

-line 210: bounding box -> Bounding box

Round 2

Reviewer 1 Report

The manuscript has been revised, and it can be published as such a form.

Minor editing of English language is required.

Reviewer 2 Report

Since all my comments were addressed, I recommend to publish the paper.